# Design, Synthesis and Biological Activities of (Thio)Urea Benzothiazole Derivatives

**DOI:** 10.3390/ijms24119488

**Published:** 2023-05-30

**Authors:** Jessica E. Mendieta-Wejebe, Martha C. Rosales-Hernández, Itzia I. Padilla-Martínez, Efrén V. García-Báez, Alejandro Cruz

**Affiliations:** 1Laboratorio de Biofísica y Biocatálisis, Sección de Estudios de Posgrado e Investigación, Escuela Superior de Medicina, Instituto Politécnico Nacional, Plan de San Luis y Salvador Díaz Mirón s/n, Casco de Santo Tomás, Ciudad de Mexico 11340, Mexico; jesmenwenj@yahoo.com (J.E.M.-W.); marcrh2002@yahoo.com.mx (M.C.R.-H.); 2Laboratorio de Química Supramolecular y Nanociencias, Unidad Profesional Interdisciplinaria de Biotecnología, Instituto Politécnico Nacional, Av. Acueducto s/n, Barrio la Laguna Ticomán, Ciudad de Mexico 07340, Mexico; ipadillamar@ipn.mx (I.I.P.-M.); efren1003@yahoo.com.mx (E.V.G.-B.)

**Keywords:** 2-aminobenzothiazoles, (thio)ureabenzothiazoles, benzothiazolsemicarbazides, benzothiazolsemicarbazones, acyliso(thio)cyanates, *N*-acyl-(thio)ureabenzothiazoles guanidinobenzothiazoles

## Abstract

(Thio)ureas ((T)Us) and benzothiazoles (BTs) each have demonstrated to have a great variety of biological activities. When these groups come together, the 2-(thio)ureabenzothizoles [(T)UBTs] are formed, improving the physicochemical as well as the biological properties, making these compounds very interesting in medicinal chemistry. Frentizole, bentaluron and methabenzthiazuron are examples of UBTs used for treatment of rheumatoid arthritis and as wood preservatives and herbicides in winter corn crops, respectively. With this antecedent, we recently reported a bibliographic review about the synthesis of this class of compounds, from the reaction of substituted 2-aminobenzothiazoles (ABTs) with iso(thio)cyanates, (thio)phosgenes, (thio)carbamoyl chlorides, 1,1’-(thio)carbonyldiimidazoles, and carbon disulfide. Herein, we prepared a bibliographic review about those features of design, chemical synthesis, and biological activities relating to (T)UBTs as potential therapeutic agents. This review is about synthetic methodologies generated from 1968 to the present day, highlighting the focus to transform (T)UBTs to compounds containing a range substituents, as illustrated with 37 schemes and 11 figures and concluded with 148 references. In this topic, the scientists dedicated to medicinal chemistry and pharmaceutical industry will find useful information for the design and synthesis of this interesting group of compounds with the aim of repurposing these compounds.

## 1. Introduction

(Thio)urea ((T)U) is a valuable functional group present in molecular structures used in several areas of chemistry and drug design. Recently, (T)U compounds have gained the attention of several researchers due to their chemical and biological properties. Due to the NH hydrogen atoms and their ability to form hydrogen bonds with the target substrates, TUs have been used in the design of molecules of biological interest. They show a broad array in their biological properties, such as anti-HIV [1], analgesic [2], antibacterial [3,4,5], antimicrobial [6,7,8,9,10,11,12,13], anticancer [14,15,16,17,18], antifungal [19,20], diuretic [21], antiviral [22,23], anticonvulsant [24], anti-thyroidal [25], herbicidal and insecticidal [26], anti-inflammatory [27], anti-acetylcholinesterase [28,29], anti-tuberculosis [30,31], antimalarial [32,33], hypoglycemic [34], CCR4 antagonists [35], and DNA-topoisomerase inhibitors [36].

Among drugs used in current medicine, some of them have a benzothiazole (BT) nucleus. BT core was found to be in natural compounds with biological activities. Since the pharmacological profiling of riluzole (6-trifluoromethoxy-benzothiazole-2-amine) became known as a clinically anticonvulsant drug in 1950 [37,38], medicinal chemists have been interested in BT derivatives. The development in biological evaluation of this heterocyclic compounds has initiated the design of novel molecules based on their mechanism of action. Nowadays, BT itself is a very important phamacophore in medicinal chemistry due to the broad pharmacological properties of its derivatives. For instance, from the period 2000–2010, only five scientific reviews were found in the literature, briefly describing important findings on synthetic methodologies and medicinal activities associated with BT compounds [39,40,41,42]. However, in the past decade, the use of BT core for drug development, has been increased rapidly. Thirty reviews were found that focused on research highlighting anticancer, antimicrobial, anticonvulsant, anti-inflammatory, antifungal, anti-oxidant, anti-tubercular, anti-malarial, anti-leishmanial, anti-Alzheimer, anti-diabetic, and other miscellaneous activities [43,44,45,46,47,48,49,50,51,52,53,54,55,56,57,58,59,60,61,62,63,64,65,66,67,68,69,70,71,72]. Some BTs are used as pharmaceuticals for the treatment of diseases such as cancer, neurodegenerative disorders, Huntington’s disease, and local brain ischemia. In the same period, an article review for the period from 2015 to 2020, relating to patents of pharmacological activities of BTs, was reported [73]. Even from the last past year until now, thirteen reviews about this topic have been reported in the literature [74,75,76,77,78,79,80,81,82,83,84,85].

Nowadays, scientists are fighting to find drugs against dangerous viruses for human health. For instance, human immunodeficiency, poliovirus, and coronavirus has destroyed many lives. Scientists have found many antiviral drugs per type of virus. From them, BT derivatives have been proved to act as good antiviral agents. Researchers are continually working on the use of BT moiety to obtain best BT derivatives to eventually be used as antiviral agents. Recently, two article reviews related to the synthesis, structure–activity relationship, and several methods for evaluating the antiviral activity against specific viruses of BT derivatives were reported [86,87]. For instance, the frentizole drug (Figure 1), a nontoxic UBT used to treat the rheumatoid arthritis and systemic lupus erythematosus were studied for the immunosuppressive and super-immunosuppressive dose levels on the resistance of the mice to viral infections. It was found that frentizol reduced the herpes simplex and influence virus. Additionally, it is known that various (T)UBTs inhibit DNA topoisomerase or HIV reverse transcriptase (e.g., **I**, Figure 1) [88,89,90]. Additionally, bentaluron and methabenzthiazuron are fungicides used as a wood preservative and herbicide in winter corn crops, respectively [91,92]. In addition, among the drugs discovered in the recent years, several (T)UBTs are potential drugs for the Alzheimer′s disease treatment [93,94,95,96,97,98].

The broad spectrum of biological activity of (T)U as well as BT derivatives encouraged us to review the methods of synthesis of (T)UBTs and their biological activity studies. In 2022, we published a literature review relating to the research progress on the synthesis and biological importance of (T)UBTs. In that work, we summarized the methodologies to synthetize (T)UBTs from the condensation of substituted 2ABTs with iso(thio)cyanates, (thio)phosgenes, 1,10-(thio)carbonyldiimidazoles (thio)carbodiimides, (T)CDIs, (thio)carbamoyl chlorides, and carbon disulfide as starting materials, covering the last 40 years [99]. In view of the fact that the synthesis and biological activity of new 2-(T)UBT derivatives have been considerably increased, and in continuation with our research in this work, we prepared a bibliographic review about these kinds of compounds; this work relates to the synthetic procedures to access (T)UBT derivatives, starting from (T)UBT as starting materials and their analysis regarding to their biological activities, from 1968 to the present day.

The work was divided into reaction methodologies starting with (un)substituted-2ABT to afford the respective (T)UBT, which were transformed into (1) (T)UBT derivatives, (2) BT-semi(thio)carbazide derivatives, (3) *N*-acyl-(T)UBT derivatives, and (4) GBT derivatives.

## 2. (T)UBT Derivatives

*N*-(4,6-disubstituted-BT-2-yl)-*N′*-(β-bromopropionyl)ureas **1a**–**g** or the N-(6-disubstituted-BT-2-yl)- *N′*-(acryloyl)ureas **2a**,**b** were prepared from the reaction between 4,6-disubstituted-2ABT and β-bromopropionyl isocyanate or acryloylisothiocyanate, respectively, Figure 1 [100]. The ureas **1d**,**e** and **2a**,**b** were cyclicized to 1-(6-disubstituted-benzothiazol-2-yl)dihydrouracils **3a**,**b**, respectively. All compounds were analyzed in vitro against antibacterial, antifungal, and antiprotozoal effects. The minimal inhibition concentration of 6-thiocyanate-*N*-β-bromopropionyl-UBT **1g** was 50 μg/mL for all tested organisms. However, the minimal inhibition concentrations of dihydrouracil-BT **3a** and **3b** were higher for all tested organisms.

The *N*-(4-methyl-BT-2-yl)-*N′*-(BI-*N*-sulfonyl)urea **4** was prepared from the reaction of BI-*N*-sulfonyl chloride and 4-methyl-UBT as starting materials, Figure 2 [101]. The antitumor activity of this compound showed GI_50_, TGI, and LC_50_ values of 25.1, 77.5, and 93.3 μM, respectively. Compound **4** had a broad-spectrum antitumor activity and selective activity to single cell lines. Distinctive activities were showed compared to that of sulofenur against EKVX non-small lung cancer, RPMI-8226 leukemia, OVCAR-4 ovarian cancer, PC-3 prostate cancer, CAKI-1 renal cancer, MDA-MB-435, and T-47D breast cancer with GI_50_ values of 1.7, 21.5, 25.9, 28.7, 15.9, 27.9, and 15.1 μM, respectively.

Nine *N*-(4/6-substituted-BT-2-yl)-*N′*-(phenyl)thioureas **5**, obtained from 4/6-substituted 2ABTs and phenylisothiocyanate in refluxing ethanol, were reacted with malonic acid in presence of acetyl chloride to synthesize 1-(4/6-substituted-BT-2-yl)-3-phenyl-2-thiobarbituric acids **6a**–**i**, Figure 3 [102]. Compounds **5** and **6** were screened for their entomological and antibacterial activities. They were tested for antibacterial activity at 200 μg/mL and 100 μ/mL in DMF against Gram-(+) and Gram-(−) bacteria, using streptomycin and ceftazidime as reference drugs. Compounds **6a**, **6f**, and **6g** showed broad-spectrum activities, but they were less active than the reference drugs. The same compounds had potent antiulcer, anti-inflammatory, antitumor, antifeedant, and acaricidal activities against *Spodoptera litura* and *Tetranychus urticae*, respectively. These compounds was proposed to be better used in drug development for bacterial infections, antifeedant, and acaricidal activities in the future as well.

*N*-(6-cyano-BT-2-yl)-*N′*-(*p*-MeObenzyl)urea **7**, obtained from 6-cyano-2ABT and the corresponding benzylisocyanate, was utilized in the generation of *N*-(6-tetrazolyl-BT-2-yl)-*N′*-(*p*-methoxy-benzyl)urea **8** in 91% yield using microwave irradiation, Figure 4 [103]. The arylureas **7** and **8** were evaluated for their inhibition of GSK-3 activity. Compound **8** was found to reduce in vitro the GSK-3β activity beneath 57% at a concentration of 1.0 μM. The structure–activity relationship of the library provided the rationale for the synthesis of UBT **8** (IC_50_ = 140 nM), which displayed more than a two-fold increase in activity compared with the standard reference AR-A014418 (IC_50_ = 330 nM).

A series of twelve *N*-(4/6-substituted-BT-2-yl)-*N′*-(*p*-substituted-aryl)ureas **9a**–**l** was synthetized for its chemoselective oxidative cyclization with 1,3-di-*n*-butylimidazolium tribromide [bbim-Br_3_] to yield the *N*-bis-BTs **10a**–**l**, Figure 5 [91]. The cytotoxic activity against a mouse melanoma cell line (B16-F10) and two human monocytic cell lines (U937, THP-1) of compounds **9** and **10** were evaluated. The compounds **9b**, **9e**, **9f**, **9k**, **9c**, and **10h** were the most active based on their IC_50_ values. Among them, compound **9e** (16.23 μM) showed the more antiproliferative activity on U937 cells compared with the reference, etoposide (17.94 μM). Nevertheless, these compounds showed less cytotoxicity towards THP-1 cells.

*N*-(6-Methoxy-BT-2-yl)-*N′-(3*-methoxy-phenyl)(thio)ureas **11a**,**b** were synthesized via amide coupling, starting from 6-methoxy-2ABT and 3-methoxyphenylisocyanate or 3-methoxyphenylisothiocyanate, respectively, Figure 6 [104]. Boron tribromide was used for the methoxy group cleavage to yield 6-hydroxy,N-*m*-hydroxyphenyl-T(U)BTs **12a**,**b**. The extension of the bridge to three units resulted in the inactive urea **12a**, but interestingly, the thiourea **12b** showed a moderate inhibitory activity on human 17β-HSD1 cancer cell line.

The *N*-(5-Benzyloxy, 7-Bromo-BT-2-yl)-*N′*-(ethyl)urea **13** was obtained in 75% yield from the reaction of 5-BnO,7-Br-2ABT with ethylisocyanate, Figure 7 [105]. The pyridine ring was coupled to the C-7 of compound **13** to afford compound **14**. Compound **14** was treated with methane sulfonic acid to yield the alcohol **15**, which was converted to the triflate **16**, and a Miyaura-borylation yielded the intermediate **17**. Suzuki cross-coupling of boronic acid **17** with the corresponding 2-substituted-5-bromo-pyrimidine **A**, followed by saponification, yielded the target products **18a**–**d**. All these compounds showed bacterial DNA gyrase and topoisomerase IV inhibition. Antibacterial properties against DNA gyrase ATPase and potent activity against *Staphylococcus aureus*, *Enterococcus faecalis*, *Streptococcus pyogenes* and *Haemophilus influenzae* were tested. To increase drug-likeness analogues with a α-substituent to the carboxylic acid group, compounds **19a**–**h** and **20a**–**d** were designed and synthesized by using the same procedures. Compounds **19a** and **19b** had IC_50_ values of 0.012 and 0.008 μg/mL for *S*. *aureus* topoisomerase IV, comparable to their *S*. *aureus* DNA gyrase ATPase IC_50_ values. Additionally, compounds **19a** and **19b** showed specificity for bacterial topoisomerases, with no inhibition of human topoisomerase II.

The same procedure was used to synthesize a series of dual-targeting, alcohol- and diol-containing *N*-(5-pyrimidin-5-yl-7-pyridie-2-yl-BT-2-yl)-*N′*-(ethyl)ureas **21**, **22**, and **23**, which showed superior antibacterial activity and drug-like properties, Figure 2 [106]. To improve the pharmacokinetic profile, the *N*-ethyl-UBTs **21a**–**c** were synthesized. The SAR study of this series showed that alcohol-containing compound had potent antibacterial activity against a primary panel of pathogenic bacteria compared to the carboxylic acids **19**, Figure 7. These compounds displayed potent antibacterial activity with minimum inhibitory concentration (MIC) against *S*. *aureus* 29,213 (0.03–0.06 μg/mL), *S*. *pyogenes* 0.06–0.12 μg/mL), and *H*. *influenza* 49,247a 0.25–1 μg/mL). Additionally, **21a**–**c** displayed a significative inhibitory activity against purified *S. aureus* T173 GyrB (IC_50_ 0.25 μg/mL all cases) and *S*. *pyogenes* ParE (IC_50_ to 0.5–1 μg/mL). Compounds **21a,d**–**m** showed potency in the nanomolar range against DNA gyrase ATPase in a malachite green assay. Bulky groups at the tertiary, *pseudo*benzylic position of the pyrimidine ring resulted in a loss of activity, especially against the ParE mutant strain of *S*. *pyogenes* (**21d** diethyl, IC_50_ 8 μg/mL; **21f** cyclohexyl ring system, IC_50_ > 16 μg/mL). However, increased on-target ATPase inhibition was observed when a heteroatom is present into the alicyclic 6-membered ring (**21g**–**j**), IC_50s_ *S*. *aureus* GryB 0.25–8 μg/mL; *S*. *pyogenes* ParE 1–8 μg/mL, suggesting a conformational role, possibly a flattening of the ring. Reduced antibacterial activity against *S*. *aureus* 29213 (MIC > 16 μg/mL), *S*. *pyogenes* 51339 (MIC 8 μg/mL; IC_50_ > 16 μg/mL) and *H*. *influenza* 49247a (MIC 16 μg/mL) was observed when the ring was an azetidine (**21l**). However, strong potency (MIC 0.06–0.5 μg/mL) across the entire tested bacterial strains (with IC_50s_ values of 0.5 and 2 μg/mL against *S*. *aureus* GryB and *S*. *pyogenes* ParE, respectively) was observed for the oxetane analogue **21m**. Longer-chain alkyl R^1^ and small cycloalkyl groups (**21n**–**q**) showed no effect on the potency of the serie toward the enzyme. However, the bulky *t*-butyl moiety (**21p**) gave a considerable drop in activity against *Haemophilus influenza* and against *S. aereus* in the presence of serum. The morpholine ring (**21s**) and diol (**21r**), designed to increase hydrophilicity, were well tolerated. Prompted by the possible steric constraint around the pyrimidine ring, the secondary alcohols **22a**–**k** and diol-containing molecules **23a**–**f** (Figure 2) were explored to increase the solubility. Compounds **22a**–**f**, with increased hydrophilicity and reduced protein binding, showed potent Gram-(+) antibacterial activity. Compounds **22a**–**c** increased the unbound fraction from 9% to 19%. However, this highlight was offset by an increase in MIC. Compounds without azetidine moieties **22i**–**k** increased this property. Compounds **23a**–**f**, with small groups, did not significantly hinder the antibacterial activity. However, diol **23d** showed some intolerance of larger rings.

The compound (*Z*)-ethyl-2-((*Z*)-2-(BT-2-ylimino)-4-oxo-3-phenylthiazolidin-5-ylidene)acetate **24** was designed to be synthesized under catalyst-free conditions, Figure 8 [107]. Compound **24** was shown to be a receptor based on an internal charge transfer mechanism with the BT unit. The density functional theory calculations (DFT) complemented these results. The fluorescent emission of compound **24** in CH_3_OH/H_2_O (50:50 *v*/*v*) was quenched in the presence of Cu^2+^ and Hg^2+^, but not in the presence of other tested metal ions. Compound **24** was complexed in a 1:1 stoichiometry with Cu^2+^ and Hg^2+^ with detection limits of 0.36 and 2.49 mM, respectively. In addition, the interactions of **24** with aldose reductase inhibitor were investigated via the use of molecular docking studies.

*N*-(6-Bromo-BT-2-yl)-*N*′-(alkyl)ureas **25a**–**l**, obtained from the reaction of 6-bromo-2ABT with CDI and alkylamine, were successively reacted with the sulfonamide compound **26**, catalyzed by bis(pinacolato)diboron and PdCl_2_(dppf), to afford *N*-[6-(2,3-disubstituted-pyridin-5-yl)-BT-2-yl)-*N′*-(alkyl)ureas **27a**–**l** and **28m**–**r** in 31–52% yield, Figure 9 [108]. The antiproliferative activities of all synthesized compounds were tested in vitro against human colon HCT116, breast MCF-7, glioblastoma U87 MG, and adenocarcinoma A549 cell lines. The compounds with potent antiproliferative activity were tested for their acute oral toxicity and inhibitory activity against PI3Ks and mTORC1 pathways. The compound connected at 2-(dialkylamino)-N-ethylurea moiety at the 2-position of BT retained the antiproliferative and inhibitory activities against phosphatidylinositol-3-kinase (PI3K) and mamalian target of rapamicyn (mTOR). Additionally, their acute oral toxicity was reduced dramatically. Moreover, compound **27f** inhibited tumor growth in a mice S180 homograft model. These results suggested that 1-(2-dialkyl-aminoethyl)-3-(6-(2-methoxy-3-sulfonylaminopyridin-5-yl)-UBTs could be used as potent PI3K inhibitors and anticancer agents with low toxicity. Secondly, compounds **27a**–**l** showed potent antiproliferative activities against the four cancer cell lines, and the activities of most compounds **27** were near to that of positive controls. These data suggested that alkylurea moiety was tolerable at the 2-position of BT. It was deduced that the activity of compounds **27** is related to the substituent at the 2- and 3-positions of the pyridine ring as well as to the 1-position of urea moiety. The factor that compound **27f** (0.3-0.45 μM) was more potent than compound **27b** (0.71–5.26 μM), against four cancer lines, indicates that a methoxy group at the 2-position of pyridine ring in that compound may enhance the antiproliferative activity. Activities of compounds **27a**–**l** were compared with that of compounds **28m**–**q** with different substituents at the 3-position of the pyridine ring, represented in Figure 3. Thirdly, the compounds containing the aryl-sulfonamino group at the 3-position of the pyridine ring, such as 4-methylphenylsulfonamino **27l** (0.17–0.36 μM), exhibited better potent activity than those containing the cyclopropylsulfonamino group **28o** (0.64–1.92 μM) and the cyano group **28n** (2.30–8.38 μM) at the 3-position of the pyridine ring. If there was not a substituent at the 3-position of pyridine ring **28m** (5.16–20 μM), the activity dramatically dropped. Fourthly, to investigate the SAR of compound **27**, a range of selected amines were attached to the 1-position of urea moiety. The data indicate that the methyl urea **27d** and cyclopropyl urea **27e** showed a drop in their cell-based activity against the four cancer lines compared with the standard compound. However, 2-(*N,N*-disubstituted amino)-*N*-ethylureas **27f**–**l** displayed similar to enhanced antiproliferative activity against the four cancer lines compared with the positive controls. Fifthly, the replacement of BT core in compound **27g** with thiazolo [5,4-b]pyridine moiety afforded compound **28r**, which displayed an enhanced activity against HCT-116 (0.52 to 0.25 μM) and MCF-7 (0.75 to 0.26 μM) cells and a close activity against U87 MG (0.49 to 0.48 μM) and A549 (0.43 to 0.49 μM) cells. These results revealed that compound **28r** is sensitive to PI3K mutant cells. In the cell-based activity, the IC_50_ values of compound **27f** are near to that of the PI3K and mTOR dual inhibitor of the reference compound. In this sense, compound **27f** was further investigated.

To compare the activities of compounds with different substituents at the 3-position of the pyridine ring, the *N*-[6-(5-substituted,6-methoxy-pyridin-3-yl)-BT-2-yl]-*N*′-[2-(morpholin-4-yl-)ethyl]urea **28m**–**q** were synthesized by using the same procedure, Figure 3.

Fifteen *N*-(BT-2-yl)-*N′*-(aryl)thioureas **29a**–**g** were synthesized from the corresponding arylisothiocianate, which underwent cyclization to 2-(*N*-thiazolidin-2-ene-4-one)-ABTs **30a**–**g** via treatment with chloroacetamide in acetonitrile, Figure 10 [109]. All compounds were evaluated with respect to their cytotoxicity against human leukaemia/lymphoma and solid tumor-derived cell lines and of their antiviral activity against HIV-1 and representatives of ssRNA and dsDNA viruses. Compound **29e** showed good activity against HIV-1 wild type and variants carrying clinically relevant mutations. A colorimetric enzyme immunoassay clarified its mode of action as a non-nucleoside inhibitor of the reverse transcriptase. The cyclization of compounds **29a**–**g** resulted in a whole decrease in their biological activities. Indeed, the anti-HIV activity of **29e** is completely lost in the derivative **30e**. Only **30b** remains cytotoxic even at a lower value (CC_50_ from 1.6 to 9 μm). Compound **30b** also seems to be less cytotoxic in the other antiproliferative assays, when compared with others and with its linear analogue **29b**.

The 6-nitro-2ABT was reacted with ethylisocyanate to afford the *N*-(6-nitro-BT-2-yl)-*N*′-(Ethyl)urea **31**, and the NO_2_ group was reduced to afford the *N*-(6-amino-BT-2-yl)-*N′*-(ethyl)urea **32**, Figure 11 [110]. Compound **32** was used to synthesize a series of *N*-(6-substituted-BT-2-yl)-*N*′-(ethyl)ureas **33**–**37**. Acylation of the 6-amino group of **32** with ethyl oxalyl chloride provided the ester **33a**, which was hydrolyzed to the acid **33b**. The compound **32** was coupled with the corresponding 2-(2-aminothiazol-4-yl)acetic acid derivative to yield carbonyl derivatives **34a**–**c**. Additionally, amine **35**, produced by Boc-deprotection of **34b**, was acylated with methyl malonyl chloride to produce **36a** and with ethyl oxalyl chloride to produce **36b**, which was hydrolized to the respective carboxylic acid **37**. All compounds were evaluated for their *Escherichia coli* DNA gyrase inhibition. Among them, the UBTs **34a**–**c** and **36a** were the most potent DNA gyrase inhibitors. Compound **34b** showed an MIC of 50μM against an *E*. *coli* efflux pump-defective strain.

The treatment of 6-substituted-2ABTs with carbon disulfide in alkali media produced a dithiocarbamate as an intermediate, which was in situ reacted with dimethyl sulfate to afford 6-substituted-2-(S-methyl-dithiocarbamate)-BTs **38a**–**d** in excellent yields, Figure 12 [111,112]. Compounds **38a**–**d** were reacted with ammonia or methyl anthranilate to be transformed to the *N*-(6-substituted-BT-2-yl)urea **39a**–**d** and the 2-[3-(6-substituted-BT-2-yl)-thioureido]benzoic acid methyl ester **41a**–**d**, respectively. The cyclization reaction of TBTs **39a**–**d** and 2-bromoacetophenone, in the presence of Et_3_N, affords *N*-(6-substitued-BT-2-yl)-4-phenyl-1,3-thiazol-2(3H)-imines **40a**–**d**. Additionally, S-methyl-dithiocarbamates **38a**–**d** reacted with methyl anthranilate to afford 3-(BT-2-yl)-2-thioxo-2,3-dihydroquinazolin-4(1H)-ones **42a**–**d**, which, after methylation with dimethyl sulfate, gave compound **43**. Compounds **40a**–**d** were tested for anti-tumor activity against MCF-7, MAD-MD-231, HT-29, HeLa, Neuro-**40a**, K-562, and L-929 cell lines, and the MTT-assay revealed compound **40b** with a phenolic segment, which resulted in having the best cytotoxicity (IC_50_ = 5.94 ± 1.98 μM) containing a phenolic segment. The apoptosis assay for compound **40b**, carried out by flow cytometry, supported the results.

## 3. BT-Semi(Thio)Carbazide Derivatives

A set of 6-substituted-BT-thiosemicarbazones **45**–**48** was synthesized from the corresponding thiosemicarbazides **44** to be screened for anticonvulsant and neurotoxic properties, Figure 13 [113]. Majority of the compounds showed anticonvulsant activity against both maximal electroshock seizure (MES) and subcutaneous pentylenetetrazole (scPTZ) screens. Eight compounds showed good protection in the rat p.o. MES test at 30 mg kg^−1^. Compound **45a** resulted as the most promising one with an ED_50_ of 17.86 and 6.07 mg kg^−1^ in mice i.p. and rat p.o., respectively. Compound **45a** showed a weak ability to block the expression of fully kindled seizures. On the other hand, the 6-methyl BTyl-2-thiosemicarbazones **47a**–**g** had anticonvulsant activity in both mice i.p. and rat oral MES screen [114]. The 6-nitro-Btyl thiosemicarbazone **46a** was the most promising one with anti-MES activity in mice i.p., rat i.p., and rat p.o. evaluations (MES screen 0.5 h; 300 mg kg^−1^) (scPTZ screen 4 h; 300 mg kg^−1^). All the compounds showed lesser or no neurotoxicity compared with the standard, phenytoin. The isatinimino derivatives exhibited better activity when compared with the benzylidene or acetophenone derivatives.

In 2007, a series of 6-substituted-BT-semicarbazones **49a**–**o** were prepared in satisfactory yield from the reaction of 6-substituted-BT-semicarbazides with *p*-substituted-acetophenones or -benzophenone, Figure 14 [115]. All synthesized compounds were evaluated for their anticonvulsant, neurotoxicity, and other toxicity studies. Most compounds were active in maximal electroshock screen (MES). Compounds **49a**, **49h**, **49m**, and **49o** possessed 100% protection at both 0.5 h and 4 h time intervals, but for compound **49a**, its percentage protection decreased to 83.3% at the 4 h interval, indicating the rapid onset but shorter duration of action under MES test conditions. The compounds containing more lipophilic substituents such as methyl and chloro groups on the BT ring were more active.

The BT-thiosemicarbazones **51a**–**c** in the presence of acetic anhydride were cyclized to produce the *N*-Acetyl, *N*-(5-aryl-4,5-dihydro-1,3,4-thiadiazol-2-yl)-2ABTs **52a**–**c**, Figure 15 [116]. The in vitro activity against *Schistosoma mansoni* at three different dosage levels (10, 50, and 100 μg/mL) were screened for selected compound. The activity of BT-thiosemicarbazide **50** was proportional to its concentration, while the thiosemicarbazones **51a**–**e** showed a dramatic decrease in activity (**51a**,**c** were inactive). In addition, thiadiazole derivatives **52a**–**c** were moderately active. It was concluded that a dithiocarbamate link is required for better activity.

*N*-(6-Substituted-BT-2-yl)ureas **53**, obtained from reaction of 6-substituted-2ABTs with HOCN, were functionalized with hydrazine, then reacted with substituted benzaldehydes to afford the corresponding BT-semicarbazone, and the cyclization of the BT-semicarbazones with chloroacetylchloride afforded a series of 6-substituted-BT(*N*-4-aryl-3-chloro-azetidinyl)semicarbazides **54a**–**t**, Figure 16 [117]. Their anticonvulsant, hepatotoxic and neurotoxic properties were evaluated. All compounds were screened for their anticonvulsant activity in an i.p. MES model and were compared with the standard drug phenytoin. Compounds **54f**, **54n**, and **54p** exhibited 100% protection in the MES test. In the neurotoxicity and hepatotoxicity screening, all the compounds were devoid of toxicity at the dose of 30 mg/kg body weight. The study showed that the introduction of F or CH_3_ groups at the 6-position of BT moiety with H, OCH_3_ groups at the 3-position and OH, and OCH_3_ groups at the 4-position of the distant phenyl ring led to increased activity. The introduction of F, NO_2_, CH_3_, and OCH_3_ groups at the 6-position of the BT moiety and un-substituted distant phenyl ring showed a moderate decrease in activity. Compounds **54e**, **54i**, **54m**, **54o**, and **54q** showed their ability to prevent seizure spread with 83% protection, whereas compounds **54a**, **54b**, **54d**, **54l**, and **54t** showed 66% protection, indicating their ability to elevate the seizure threshold. Compounds **54c**, **54g**, **54h**, **54j**, **54k**, **54r**, and **54s** showed 50% protection. Thus, most compounds displayed preferential MES selectivity. In the neurotoxicity screen, all the compounds were not toxic at the dose of 30 mg/kg body weight.

The *N*-(5-methyl-BT-2-yl)thiourea, obtained from the reaction of substituted 5-Me-2ABT with ammonium thiocyanate, was used as an intermediate to be reacted with hydrazine hydrate in ethylene glycol and then with substituted acetophenones to form the respective 5-methyl-BT-thiosemicarbazone **55a**–**c**, Figure 17 [118]. All compounds were tested against antimicrobial activity by using Gram-(+) and Gram-(−) bacteria such as *S*. *aureus*, *P*. *aeruginosa*, and *E*. *coli* with ampicillin as standard compound. Derivatives showed a significant zone of inhibition against both Gram-(±) bacteria at 1 mg/mL. Then, compound **55a** showed excellent activity against *P*. *aeruginosa*, while compound **55b** showed excellent activity against *E*. *coli*, while compound **55c** showed moderate activity against all three bacteria.

Six 6-substituted-BT-semicarbazides, obtained from the reaction of 6-substituted-2ABTs with sodium cyanate in acetic acid, then hydrazine in ethanol, were used as intermediates in the reaction with phenylisothiocyanate and concentrated sulfuric acid to afford *N*-(5-aminophenyl-[1,3,4]thiadiazol-2-yl)-2ABTs **56a**–**f**, Figure 18 [119]. The compounds were screened for anticancer activity (Human Chronic Myelogenous Leukemia) using in vitro assays. All compounds exhibited from a high to moderate anticancer activities. Compounds **56c** and **56d** showed the best anticancer potential because the presence of highly electronegative and electron-donating groups.

A series of *N*-(6-chloro-BT-2-yl)-*N*-([1,2,4]-triazolo-[3,4-b]-[1,3,4]-thiadiazole)amines **59a**–**g** and *N*-(6-chloro-BT-2-yl)-*N*-(5-substituted-[1,3,4]-oxadiazol-2-yl)amines **60a**–**g** and **61** were synthesized from the intermediate 6-chloro-BT-semicarbazide **57** obtained from 6-chloro-2ABT, NaOCN, and then hydrazine, Figure 19 [120]. Their antimicrobial properties were investigated against one Gram-(+) bacteria (*Staphylococcus aureus*), three Gram-(−) bacteria (*Pseudomonas aeruginosa*, *Escherichia coli*, *Klebsiella pneumoniae*), and five fungi (*Penicillium citrinum*, *Candida albicans*, *Aspergillus flavus*, *Aspergi llus niger*, and *Monascus purpureus*) using the serial plate dilution method. All the tested compounds exhibited moderate to good antibacterial and fungal inhibition at 12.5–100 μg/mL in DMSO. The triazolo-thiadiazole-ABTs **59a**–**g** were found to be more active than 1,3,4-oxadiazole-ABTs **60a**–**g** and **61** against all pathogenic bacterial and fungal strains.

6-chloro-BT-thiosemicarbazones **62a**–**h** were synthesized by Amir et al., Figure 20 [121]. All synthesized compounds were screened against the in vivo anticonvulsant, and acute toxicity. Additionally, a 3D four-point pharmacophore measurements of the compounds was carried out to be compared with established anticonvulsants agents.

A series of 6-nitro-BT-semicarbazones **63a**–**u** and **64** were designed and synthesized, to be evaluated as inhibitors of the rat brain MAO-B isoenzyme, Figure 4 [122]. Most compounds were potent inhibitors of MAO-B, with IC_50_ values from the nanomolar to micromolar range. Molecular docking studies with AutoDock 4.2 were performed to deduce the affinity and binding mode of these inhibitors toward the MAO-B active site. The free energies of binding (ΔG) and inhibition constants (*Ki*) of the docked compounds were calculated by the Lamarckian genetic algorithm (LGA) of Auto Dock 4.2. Good correlations were obtained between the calculated and experimental values. Compound **63d** emerged as the lead MAO-B inhibitor, with top ranking in both the experimental MAO-B assay (IC_50_: 0.004 ± 0.001μΜ) and in computational docking studies (*Ki*: 1.08 μΜ). A binding mode analysis of potent inhibitors suggests that these compounds are well accommodated by the MAO-B active site through stable hydrophobic and hydrogen-bonding interactions. Interestingly, the 6-nitroBT moiety was found to be stabilized in the substrate cavity with the aryl or diaryl residues extending up into the entrance cavity of the active site. Thus, a binding site model consisting of three essential pharmacophoric features was proposed with the aim being for it to be used in the design of future MAO-B inhibitors.

On the other hand, the anticonvulsant activity of 2-amino-6-nitro-BT-semicarbazones **63** and **64** in various in vivo animal seizure models were investigated, Figure 4 [123]. MES, sscPTZ, and 6 Hz psychomotor seizure models were tested. Neurotoxicity was estimated by the rotarod test. Additionally, the compounds were assessed for their neuroprotective potential from excitotoxic insult using organotypic hippocampal slice culture neuroprotection assay. Various compounds exhibited excellent anticonvulsant activity in MES and scPTZ models compared to reference drugs, phenytoin and levetiracetam. The results of kainic acid (KA)-induced neuroprotection assay showed that compounds **63r** and **63t** were most potent with IC_50_ values of 101.00 ± 1.20 μM and 99.54 ± 1.27 μM, respectively. Both compounds attenuated KA-mediated cell death in organotypic hippocampal slice cultures. Some compounds were better antidepressants compared with the reference drug citalopram, when analyzed in a forced swim test. Since semicarbazones showed a profile resembling phenytoin, an intent was made to screen them against human neuronal sodium channel isoform (hNav1.2) by carrying out computational molecular docking using AutoDock 4.2. Compound **64c** resulted in being the lead candidate with excellent in vivo MES activity and computationally better binding affinity than the reference drug phenytoin and also exhibited neuroprotection from excitotoxic insult in KA-induced neuroprotection assay (IC_50_ = 126.80 ± 1.24 μM). However, some of the active compounds were neurotoxic at their anticonvulsant doses. Optimization studies were proposed to reduce toxicity and develop them as novel epilepsy therapeutic agents.

The synthesis of potential anthelmintic agents *N*-(6-fluoro,7-chloro-BT-2-yl)-*N*′-(2-aryl, 4-one-thiazolidin-3-yl)ureas **68a**–**e** were achieved from reaction of the corresponding 6F,7Cl-2ABT and ethyl chloroformate to afford the ethyl 6F,7Cl-BT-yl-2-carbamate **65**, which, via treatment with hydrazine hydrate in ethanol, gave the 6-fluro-7-chloro-BT-semicarbazide **66**, Figure 21 [124]. The BT-semicarbazide **66** was treated with aromatic aldehydes in absolute ethanol to yield the BT-semicarbazones **67a**–**e**. Thioglycolic acid and zinc chloride were added to the semicarbazones to afford thiazolidinone derivatives **68a**–**e**. All compounds showed good to moderate anthelmintic activity against earth worms *Perituma posthuma* compared with Albendazole as the standard drug. Compounds **68c** and **68d** showed good anthelmintic activity, whereas compounds **68a**, **68b**, and **68e** displayed less or comparable anthelmintic activity.

The intermediates 6-substituted-BT-semicarbazides **69a**–**e**, obtained from reaction of 6-substituted-2ABT with phenyl chloroformate, then a hydrazinolysis, were reacted with substituted *o*-hydroxybenzaldehydes to afford a series of 6-substituted-BT-semicazone derivatives **70**–**72**, Figure 22 [125]. These synthesized compounds were screened in vitro with respect to their cytotoxic activities against five cancer cell lines (NCI-H226, SK-N-SH, HT29, MKN45, and MDA-MB-231). Most of them exhibited moderate to excellent activity against the five cell lines. Compound **71g** (procaspase-3 EC_50_ 1.42 μM) and **72b** (procaspase-3 EC_50_ 0.25 μM) showed excellent antitumor activity with IC_50_ values from 0.14 μM to 0.98 μM against all cancer cell lines, which were 1.8–8.7 times more active than the first procaspase-activating compound (PAC-1) (procaspase-3 EC_50_ 4.08 μM). The SAR analyses showed that a lipophilic group (benzyloxy or heteroaryloxy groups) at the 4-position of the 2-hydroxy phenyl ring, was beneficial to antitumor activity, and R^1^-substituents with nitrogen, which are positively charged at physiological pH, could improve antitumor activity. Additionally, it was confirmed that the steric effect of the 4-position substituent of the benzyloxy group had an influence on cytotoxicity.

On the other hand, the same procedure was used by Ma et al. to synthesize a different series of 6-substituted-BT-semicarbazones **73**–**75** for antiproliferative activities and procaspase-3 kinase activities against five cell lines (MDA-MB-231, MNK-45, NCI-H226, HT-29, and SK-N-SH), Figure 5 [126]. The BT-semicarbazone **74e** showed inhibitory activities against the five cell lines with IC_50_ and EC_50_ values from 0.4 to 0.92 μM and 0.31 μM, respectively. The SAR studies showed that the in vitro pharmacological activities of BT scaffold **74e** are due to the introduction of phenyl and benzyloxyl substitutions.

From the condensation reaction of the respective 6-substituted-BT-semicarbazide with the corresponding aldehyde or ketone derived from substituted indole derivatives, a series of 6-substituted-BT-semicarbazones with an indole-based moiety **76**–**78** were designed to be synthesized and screened for in vitro antitumor activity against four cancer cell lines (HT29, H460, A549, and MDA-MB-231), Figure 6 [127]. Most of them showed moderate to excellent activity against the four cell lines. The compounds **78a**–**zi** exhibited better selectivity against the HT29 cancer cell line. Among these derivatives, the carboxamides **78za**, showed potent antitumor activity with IC_50_ values of 0.015 μM for HT29, 0.28 μM for H460, 1.53 μM for A549, and 0.68 μM for MDA-MB-231. On the other hand, compound **78f** exhibited excellent antitumor activity with IC_50_ values of 0.024, 0.29, 0.84, and 0.88 μM against the four cancer cell lines, respectively. The SAR studies showed that compound **78d** exhibited the highest antitumor activity due to the presence of electron-withdrawing groups in the 4-position of the benzyl ring. In another work, it was found that compounds with fluoro-substituted benzyl-1*H*-indole moiety displayed more potent activity than those with phenyl moiety as compounds **78** (R = F), Figure 6 [128]. Compound **76c,** the most promising, exhibited excellent antitumor activity (IC_50_ values of 0.015, 0.28, 1.53, and 0.68 μmol/L) against the four cell lines, respectively. Mechanism studies indicated that the marked pharmacological activity of compound **76c** might be ascribed to activation of procaspase-3 (apoptosis-inducing) and cell cycle arrest, which was proposed as a lead for further structural modifications. Furthermore, a 3D-QSAR model (training set: q^2^ = 0.850, r^2^ = 0.987, test set: r^2^ = 0.811) was built to provide a comprehensive guide for further structural modification and optimization.

A series of seven 6-chloro-BT-semicarbazones **79a**–**g** were synthesized by reacting the 6-chloro-BT-semicarbazide with aryl aldehydes in refluxing ethanol. These compounds were condensed with thioglycolic acid to suffer ring closure to give compounds **80a**–**g**, Figure 23 [129]. On the other hand, compounds **81a**–**g** were synthesized by reacting compounds **79a**–**g** with chloroacetyl chloride. All compounds were tested against one Gram-(+) bacteria (*Staphylococcus aureus*), three Gram-(−) bacteria (*Pseudomonas aeruginosa*, *Klebsiella pneumonia*, and *Escherichia coli*), and five fungi (*Aspergillus niger*, *Aspergillus flavus*, *Candida albicans*, *Penicillium citrinum*, and *Monascus purpureus*) using the serial plate dilution method. All the synthesized compounds showed from moderate to good antibacterial and antifungal inhibition (12.5–200 μg/mL in DMSO) compared with the standard drug Ofloxacin. The azetidin-2-ones **81a**–**g** were more active than thiazolidin-4-ones **80a**–**g** against all pathogenic bacterial and fungal strains. On the other hand, the in vivo anticonvulsant screening of these compounds revealed that **80c**, **80g**, and **80c** had promising anticonvulsant activities without any neurotoxicity [130]. The evaluation of selected compounds for their in vitro GABA AT inhibition showed that compound **80c** (IC_50_ 15.26 μM) exhibited excellent activity compared with the vigabatrin as standard drug (IC_50_ 39.72 μM), suggesting the potential of these BT derivatives as new anticonvulsants.

A series of 6-substituted-BT-semicarbazone compounds **82a**–**l** was synthesized from the reaction of 6-substituted-BT-semicarbazide with the corresponding aldehyde or ketone for their in vitro evaluation in antiproliferative activity against procaspase-3 over-expression in cancer cell line U937 and a procaspase-3 no-expression in cancer cell line MCF-7, to rule out off-target effects, Figure 24 [131]. Biological evaluation identified a series of BTs bearing a pyridine-semicarbazone moiety, **82j** and **82k**, with promising antiproliferative activity and significant selectivity. Mechanism studies showed that compounds **82j** and **82k** could induce apoptosis of cancer cells by activating procaspase-3 to caspase-3, and compound **82k** showed the strongest procaspase-3 activation activity. SAR studies revealed that the presence of BT and an N,N,O-donor set is crucial for the antiproliferative activity and selectivity, and reducing the electron density of the N,N,O-donor set resulted in a dramatic decline in both. Furthermore, toxicity evaluation (zebrafish) in vivo and metabolic stability studies (human, rat, and mouse liver microsomes) were achieved to provide reliable guidance for further PK/PD studies.

Twelve 6/4-substituted-BT-semicarbazone-bis-sulfoxides **84a**–**l**, as potential antibacterial agents, were designed and synthesized from the reaction of 6/4-substituted-BT-semicarbazides with carbon disulfide and then with the corresponding alkyl iodide to afford the intermediate dithioalkylcarboimidate derivative **83a**–**l**, which were further oxidized in the presence of sodium tungstate dihydrate, Figure 25 [132]. The results of the bioactivity assays showed that many compounds had significant in vitro inhibitory effects against *Xanthomonas oryzae* pv. *oryzae* (*Xoo*) and *Xanthomonas citri* pv. *citri* (*Xac*). Compound **84g** had the best in vitro antibacterial activity against *Xoo* at a half-maximal effective concentration value of 11.4 μg/mL, which was superior to those of thiadiazole copper (TDC) and bismerthiazol (BMT). Compared with TDC and BMT, compound **84g** was more effective in in vivo controlling rice bacterial leaf blight with curative and protection activities of 42.5% and 40.3%, respectively. Additionally, compound **84g** influenced biofilm formation, inhibiting extracellular polysaccharide production, and ultimately, it reduced the pathogenicity of *Xoo*. All the results showed that bis-sulfoxides bearing acylhydrazone and BT moieties could be used as a base for the development of small-molecule pesticides with good antibacterial activity.

A series of *N*-(BT-2-yl)-*N′*-(substituted)(thio)ureas **86**–**89** was obtained to be tested for their in vitro cytotoxicity against MCF-7 breast cancer cells, Figure 26 [89]. The *N*-(BT-2-yl)-*N′*-(morpholin)urea **86a** showed the highest cytotoxic activity. A docked pose of **86a** bound to the G-quadruplex of the human telomere DNA active site using the Molecular Operating Environment (MOE) module was obtained. Moreover, all compounds were screened for their antimicrobial activity against *Mycobacterium tuberculosis* H_37_R_v_, *E*. *coli*, *S*. *aureus*, and *C*. *albicans*. Compound **86a** showed the best activity against *M*. *tuberculosis* H_37_R_v_ reducing the growth of MCF-7 cells to 76% at 10 μM, while other compounds were equipotent with ampicillin (12.5 and 25 μg mL^−1^) against *S*. *aureus* (**88b**) and *E*. *coli* (**89a**), respectively.

## 4. *N*-acyltUBT Derivatives

The treatment of 6/5-substituted-2ABTs with benzoyl isocyanates in acetone, afforded the *N*-(6/5-substituted-BT-2-yl)-*N′*-(benzoyl)thioureas **90a**–**k**, Figure 27 [133]. All compounds were tested in vitro for their antimicrobial activity against Gram-(+) and Gram-(−) bacteria. They exhibited moderate to potent activity, as compared to the standard drugs. However, the compounds **90a**–**k** were less bioactive than the respective substituted-2ABTs. Compound **90a** showed high activity against *E*. *coli* (35 mm of inhibition diameter at 2.0 mg/mL).

Rana et al. prepared a series of *N*-(6-substituted-BT-2-yl)-*N′*-(benzoyl)thioureas **91a**–**t** in 61–88% yield from 6-substituted-2ABTs and substituted benzoyl isothiocyanates, Figure 7 [134]: All compounds were evaluated for their neurotoxicity, CNS depressant effects, anticonvulsant activity, and other toxicity studies. With exception of compounds **91m** and **91q**–**s**, all compounds showed protection in the animal models of seizures. Compounds **91b, 91i**, and **91p** emerged as non-neurotoxic anticonvulsants and can be demanded to detect compounds possessing activities against generalized tonic-clonic (grand mal) and generalized absence (petit mal) seizures, respectively. None of the compounds showed neurotoxicity or liver toxicity. In a CNS depressant study, most compounds decreased the immobility time. For CNS depressant and anticonvulsant activity, the dose used was 30 mg/kg; however, the mechanism of action involved for both the activities, could not be proposed. However, further studies were suggested.

*N*-(BT-2-yl)-*N*′-(benzoyl)thioureas **90a** and **92** were obtained from reaction of 2ABT with the corresponding in situ-generated benzoylthiocyanate, Figure 8 [135,136]. The *N*-benzoyl-TBTs structures were analyzed by X-ray diffraction analysis. The thiourea moiety was found to be in the thioamide form. The molecules of *N*-benzoyl-TBTs were found to be stabilized by the C—H---S and C—H---O intermolecular hydrogen bonds. Additionally, a pseudo-S(6) planar ring with an intramolecular N—H---O hydrogen bonding was formed with dihedral angles of 11.23 and 11.91 with the BT ring system and the phenyl ring, respectively.

On the other hand, Siddiqui et al. refluxed *N*-(6-substituted-BT-2-yl)-*N*′-(benzoyl)thioureas **93a**–**t** in *n*-butanol to be cyclized to the phenyl-2H-[1,3,5]triazino[2,1-b][1,3]BT-2-thiones **94a**–**t**, Figure 28 [137]. All compounds were evaluated for their anticonvulsant, anti-nociceptive, hepatotoxic, and neurotoxic properties. Phenytoin was used as standard drug in the evaluation of their anticonvulsant activity in a mouse seizure model. Compounds **94a**, **94c**, **94f**, and **94l** showed total protection from 0.5 h to 4 h periods. None of the selected compounds showed any neurotoxicity or hepatotoxicity. Additionally, all compound**s** were tested for their anti-nociceptive activity by a thermal stimulus technique compared with diclofenac as the standard drug. Compounds **94o**, **94q**, and **94t** resulted in having highly potent analgesic activity with *p* < 0.01.

Some *N*-(substituted-BT-2-yl)-*N*′-(purinoacetyl)thioureas **95a**–**j** were synthesized from the reaction of the in situ-generated 2-((2-amino-9-phenylsulfonyl)-9H-purin-6-yl)amino) acetyl isothiocyanate with substituted-2ABTs. Their in vitro antimicrobial activity against Gram-(+) and Gram-(−) strains and antifungal strains was evaluated using a micro dilution procedure, Figure 29 [138]. All compounds proved to be effective (MIC values). Among them, compounds **95a** (0.17 mM) and **95d** (0.18 mM) exhibited very good activity against *Kleb*, and **95e** (0.18 mM) exhibited very good activity against *B*. *subtilis* and *A*. *niger*. Compound **95e** showed MIC value less than the standard drugs Ciprofloxacin (0.25 mM), Ampicillin (0.27 mM), and Fluconazole (0.35 mM). Electron-donating groups, such as halogen and methoxy groups, on the BT ring influenced the antibacterial and antifungal activity in comparison with the electron-withdrawing ones. Based on these results, it was proposed that the thiourea group connection between the BT and purine rings seems very important for the antimicrobial effect.

A series of twenty *N*-(5-substituted-BT-2-yl)-*N*′-acyl)thioureas **96** was efficiently synthesized and evaluated for antimicrobial and anti-proliferative activities against cancer cells, Figure 30 [7]. These compounds exhibited a broad spectrum of activity against the tested microorganisms and showed higher activity against fungi than bacteria. Compounds **96ba**, **96bb**, **96bc**, **96bd**, and **96bd** showed the greatest antimicrobial activity. Preliminary SAR studies showed that electronic factors in BT rings had a great effect on the antimicrobial activity. In preliminary MTT cytotoxicity studies, the compounds **96db**, **96cb**, and **96de** resulted in being the most potent. In MCF-7 and HeLa cells, the range of the IC_50_ values was 18–26 μM and 38–46 μM, respectively.

A series of thirty *N*-(substituted-BT-2-yl)-*N*′-[2-methyl-4-oxoquinazolinethioyl-3(4H)]thioureas, **98**–**100a**–**j**, was designed and synthesized. Their anticonvulsant activity in the seizure models *sc*PTZ, 6-Hz, and MES were evaluated, Figure 31 [139]. *N*-(substituted-BT-2-yl)-*N*′-(thiocarbamidoyl)ureas **97a**–**j**, obtained from reaction of UBTs with ammonium thiocyanate were reacted with 4-substituted-2-acetamidobenzoic acid to afford the compounds **98**–**100a**–**j**. Compounds **100a** (PI > 3.4; *sc*PTZ) and **100f** (PI > 26; MES > 4.2; *sc*PTZ) exhibited comparable to higher activities with phenytoin (PI > 22; MES) and ethosuximide (PI > 3.0; *sc*PTZ) as the reference drug. Additionally, they showed anticonvulsant activity against (S)-2-amino-3-(3-hydroxyl-5-methyl-4-isoxazolyl) propionic acid (AMPA)- and γ-amino butyric acid (GABA)-induced seizures. Compounds **100b** and **100f** could be provided to have activity at multiple receptors; thus, they were proposed as templates for future design, modification, and investigation to obtain more active analogs. However, further studies were proposed to precisely determine the mechanism and pharmacokinetic parameters.

*N*-(BT-2-yl)-*N*′-(benzoyl)thioureas **101a**–**i**, prepared from the reaction of 2ABT with the respective benzoyl isothiocyanate in refluxing acetone for 6 h, were converted in a gold(I)-mediated intramolecular trans-amidation via dethiocyanation to 2-(substituted-benzamide)-BTs **102a**–**i**, Figure 32 [140]. The density functional theory calculations support the mechanism proposed in the transformation.

## 5. Guanidine Derivatives

Since 1968, TBTs have been used as intermediates to synthetize BT-2-yl-guanidines (GBTs); for instance, *N*-(substituted-BT-2-yl)-*N*′-(*p*-tolyl)thioureas **103a**–**k**, synthesized from reaction of substituted 2ABTs with *p*-tolylthiocyanate, were reacted with the corresponding ammonia, methylamine, or ethylamine in the presence of yellow lead oxide as the desulfurating agent to afford twenty-six *N-*(substituted-BT-2-yl)-*N*′-(*p*-tolyl)-*N*″-(alkyl)guanidines **104**–**106**, Figure 33 [141]. Several of these were evaluated for their antibacterial and antitubercular activity against *M*. *tuberculosis* (H_37_Rv). All the tested compounds were inactive as antibacterials. However, compounds **104i** and **105c** were the most active as antituberculars.

On the other hand, several *N*-(substituted-BT-2-yl)-*N*′-(*p*-chlorophenyl)-*N*″-(alkyl)guanidines **107**–**111** were synthesized in 50–90% yield, Figure 9. Compounds were screened for antitubercular and antibacterial activities [142]. The tested compounds were inactive in vitro at MIC = 80 μg/mL against *M. tuberculosis* (H_37_R_V_). All compounds were screened with respect to their antibacterial activity against *S*. *aureus*, *B*. *subtilis*, *S*. *typhi*, *A*. *tumefaciens*, and *E*. *coli*. Compounds **107f** and **109f** resulted in being the most active against *B*. *subtilis* and **107f**, **107g**, **108f**, **109f**, **109k**, and **110i** against *S*. *aureus* at 6.25 μg/mL. All compounds resulted in being inactive against *A*. *tumefaciens*, *S*. *typhi*, and *E*. *coli* at a maximum concentration of 100 μg/mL.

In 1971, the same author synthesized a series of *N*-(substituted-BT-2-yl)-*N′*-(benzyl)guanidines **112a**–**k** using the same procedure, Figure 10 [143].

In view of the antiprotozoal and antifungal activity of (substituted) benzothiazolyl guanidines, the same procedure was used to synthetize a series of twenty two *N*-*p*-bromophenyl-*N*′-(substituted)-benzothiazol-2-yl-*N*″-(*n*-propyl and *n*-butyl) guanidines **113a**-**k** and **114a**-**k** Figure 11 [144].

Four *N*-(BT-2-yl)-*N*′-(aryl)-*N*″-(amino)guanidines **115a**–**d** were synthesized from the amination-desulfhydration of *N*-(BT-2-yl)-*N*′-(aryl)thioureas, Figure 34 [145]. These compounds were tested for their antimicrobial activity by the standard disc diffusion method. The activities were compared with that of standard strain of *Escherichia coli*, NCTC 10418. From the sensitive aminoguanidines, their MIC was further obtained.

*N*-(6-substituted-BT-2-yl)thioureas **116a**–**f**, obtained from the reaction of 6-substituted-2ABTs with saturated ammoniumthiocynate, were treated with β-alanine in the presence of CuSO_4_.5H_2_O as the desulfurating agent to afford the *N*-(6-substituted-BT-2-yl)-*N*′-(2-carboxy-ethanoyl)guanidines **117a**–**f**, Figure 35 [146]: Compounds **117a**–**e** showed promising cytotoxicity against the human cervical cancer cell line (HeLa); however, compound **117f** exhibited the most potent cytotoxicity. Compounds **116e** and **117b** were considered to be the most active antimicrobials, with a broad spectrum against both Gram-(+) and Gram-(−) bacteria.

Twelve acetylated *N*-(6-substituted-BT-2-yl)-*N*′-(sugar)-*N*″-(substituted)guanidines **119**–**121** were obtained in 48–77% yield from the reaction of an alkyl or aryl amine with *N*-(6-substituted-BT-2-yl)-*N*′-(sugar)thioureas **118** in the presence of HgCl_2_, Figure 36 [147]. The removal of protecting groups afforded guanidines **122**–**124** in 76–95% yield. The 6-substituted-sugar-TBTs **118** were synthesized in 52–58% yield by the reaction of 6-substituted-2ABTs with the sugar thiocyanate derivative in anhydrous pyridine. Some of the synthesized guanidines displayed anti-influenza activity.

A series of *N*-(6-trifluoromethoxy-BT-2-yl)-*N*′-(alkyl)thioureas **125a**–**d** were synthetized from 6-trifluoromethoxy-2ABT with the corresponding alkylisothiocyanate. Among them, TBTs **125a** and **125b** were reacted with methyl iodide in acetone and then with gaseous ammonia to afford the respective guanidine **126a** and **126b**, Figure 37 [148]. All compounds were tested by an in vitro protocol of ischemia/reperfusion injury, but only compounds **125a**–**d** showed a significant reduction in neuronal injury. The anti-oxidant properties of compounds **125a**–**d** were shown to be gifted with a direct ROS scavenging activity. Compounds **125b** and **125d** were tested for their activity on voltage-dependent Na^+^ and Ca^2+^ currents in neurons from rat piriform cortex. At 50 μM, compound **125b** inhibited the transient Na^+^ current to a much smaller grade than rilusole, while **125d** was near ineffective.

## 6. Conclusions

A bibliographic review about aspects of design, chemical synthesis, and biological activities related with (T)UBTs derivatives as potential therapeutic agents is herein presented.

Synthetic methodologies from 1968 to the present day, highlighting the approaches to transform (T)UBTs to afford derivative compounds with a variety of substituents, are illustrated with 37 Schemes and 11 Figures and are concluded with 148 references.

The work relates to recent reaction methodologies starting with (un)substituted-2ABT to afford the (T)UBTs, which were transformed into (1) (T)UBT derivatives, (2) BT-semi(thio)carbazide derivatives, (3) N-acyl-(T)UBT derivatives, and (4) GBT derivatives.

This topic provides information regarding the utility for medicinal chemists and the pharmaceutical industry, which are dedicated to the design, synthesis, and SAR studies of these interesting compounds. The main biological activities tested were included with the aim of provide a comprehensive overview of these compounds. The reader will find the required information to judge the suitability of compounds for testing other biological activities or pharmaceutical targets.

Most compounds were analyzed with respect to their biological activities such as ubiquitin ligase inhibitor and anti-helmintic, antitubercular, anti-nociceptive, anti-convulsant, anti-influenza, anti-inflammatory, anti-bacterial, anti-oxidant, anti-proliferative of cancer, antifungal, and antimicrobial activities. Some of these compounds are promising lead molecules to be developed as potent chemotherapeutic agents.

It was observed that the principal activities of the BT derivatives were anti-bacterial, anti-microbial, and anticancer activities.

The information provided in this review shows that (T)UBT derivatives have activity at multiple receptors. On this basis, these kinds of compounds can be proposed as templates for future design, modification, and investigation to synthesize more active analogs. On the other hand, researchers have proposed further studies to precisely determine the mechanism of action and pharmacokinetic parameters.

## Data Availability

Not applicable.

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
