# Peer review of "Design, Synthesis and Biological Activities of (Thio)Urea Benzothiazole Derivatives"

_ijms, 2023, doi:10.3390/ijms24119488_

Round 1

Reviewer 1 Report

The authors provided an extensive review upon the synthesis and biological activities of (thio)ureabenzothiazole derivatives. However, due to several comments, the manuscript could not be accepted except after considering them.

1. More consistency is needed in many aspects. For example, the way  the compounds are represented in the schemes, especially for a series of compounds, whether to nominate the starting compound or not, compounds e.g. 1 or 1a-g,..

2. The English language needs proofreading to fix many errors in typing, spelling, grammatical, style, weak expressions, unnecessary capital letters, or missed capital letters. 

3. Line 46-50, concerning technical applications, it is out of the scope of the review title; biological activities. It is advisable to be omitted.

4. Line 63, ref. 52 is the year 2014, while the text in line 60 describes 2000-2010 period. Please check.

5. Line 67, "anti-tubercular" is repeated.

6. Line 89, compound III is cited. Where are I and II?

7. In fig. 1, it is "methabenthiazuron", while "bethabenthiazuron" in the abstract and introduction. The correct is "methabenzthiazuron". Please check.

8. Line 111, 1d,e are cited. Why not 1a-g as in scheme 1?

9. Scheme 1, ketone on the arrow should be acetone.

10. Scheme 6 has no 11a,b as in line 181. Consistency is needed.

11. Chemical names should be written in a more consistent and accurate style, e.g., line 191, "(5-BzO, 7-Br), N-ethylUBT" is not suitable...

12. Line 196, "bromide A" is not clear.

13. Line 197, 18a-c, while in sch. 7, it is 18a-d.

14. Line 233, azetidine 21k. Azetidine has 3C and 1N, while R and R' for this compound as in fig. 2 is (CH2)4NH.

15. Line 252, "intolerance of the large rings"; what does it mean?

16. Line 283, compound 27f  is which structure? Structures are not shown while described in text...Sch. 9 could not be interpreted, in addition it contains (2a-c) 

17. Line 297, "cyclopropylsulfonamino" while all compd 27 are arylsulfonamino as drawn in sch. 9.

18. Line 302 and 303 seem to be a copy of the original paper which is not suitable (We, sch. 2, table 1).

19. Sch. 11, revise the structures of ethyl oxalyl chloride and methyl malonyl chloride. A CO is missing.

20. Line 369, where is the phenolic segment in 40b?

21. Sch. 13, compounds are described as 40-43. The correct is 45-48.

22.  "Aldehides' should be 'aldehydes". The same is for "benzaldehide". Please correct throughout the whole manuscript 

23. Line 444 and sch. 17, "ethylene glycol" is the correct.

24. Line 472 and sch. 19, compound 56 should be 57. All the following compounds should be corrected.

25. Line 509. "our" should be omitted.

26. Line 589, "figure 6" should be "5".

27. Line 613, "scheme 25"; it seems that there is a mistake in numbering the schemes.

28. Line 610, where i the electron withdrawing group?

29. Line 647, where are the three series of compound 81? In addition, their drawing in sch. 24 is impossible to understand.

30. Sch. 25, compounds 82 and 83 are interchanged.

31. Line 689, according to structure in sch. 26, 86a should be 85a.

32. Line 810, 100a-j should be 100a-i.

33. Line 826, what are ref. [178-184]?

34. Line 835, what is 16c? Also, what is 13f in line 868?

35. Line 839, fig. 9 should be cited in the text.

36. Line 855, "114" should be "(114a-d)".

37, In abstract and conclusion it is said that methodologies from 1973, while a ref. from 1968 is included.

Author Response

Thanks a lot for the reevision

Reviewer 2 Report

The manuscript ijms-2304063 devoted the actual field of the pharmaceutical science, namely benzothiazole as potential biologically active compouns and can be interested to the specialists working in this field. The author’s opinion is clear and based on a good literature data. I am personally impressed by the structure of the article, the systematization of scientific data and the sequence of its presentation. The paper fit the Journal scope and formal requirements. However, it needs major revision before publication.

To improve the quality and perception of the manuscript I would suggest paying attention to following comments:

1.     A key remark to the peer-reviewed manuscript is the use of chemical nomenclature that is not entirely acceptable for authoritative peer-reviewed journals. In my opinion, the use of chemical names such as 4,6-disubstituted,N- -bromopropionylUBTs or 4-methyl, N-BI-N-sulfonylUBT is unacceptable. Authors should make changes throughout the text of the article by analogy with similar reviews in chemical and pharmaceutical journals..

2.     In my opinion, it is worth expanding the conclusions. In the conclusions, it is worth detailing the obtained results of literature analysis, and their possible use in further research.

3.     References list should be carefully checked and journal style policy should be strictly followed (DOI, etc).

4.     There is highly recommended to use professional editing service to spell-check and improve the language of the manuscript. There are numerous grammar and orthographical errors in the manuscript, which should be corrected.

My decision is major revision.

Author Response

Thanks a lot for the revision

Reviewer 3 Report

Reviews are valuable because contains concentrated information focused on interesting problem. The evaluated manuscript  focuses on application of 2-aminobenzothiazole in synthesis of different its derivatives exhibiting wide range of biological activity. This review present variety of synthetic pathways leading to desired compounds, all of them use 2-aminobenzothiazole as a basic reactant.  The application of 2-aminobenzothiazole in synthesis of bio-active compounds is very actual, because many derivatives exhibit high activity against specific pathogens.  Synthesis of this compounds should open the further discussion followed presentation of examples.                                                        Problematic is form of schemes presentation. Is convenient the indicated by coloring atoms involved in formation of new bonds, this procedure help to understand the mechanism of discussed transformation. Using colors for other purposes in chemical schemes is unnecessary, e.g., blue nitrogen as an inert gas, and especially when in main reaction scheme R substituent is black. Authors should carefully check all schemes because the proportions of bonds length and angles differs for commonly accepted. Another weak point is putting reactants over and below the arrows. It commonly accepts that even in multiple steps reactions all particular reactants are located over arrow, a place below is reserve for description of applied conditions and eventual by-products.  In review is unnecessary present all synthesized compounds, is better focus on active one, it simplified schemes and ensures clarity of discussion course. Authors should carefully check chemical nomenclature. The name nitrile (e.g., page 5 line 156) should be change to cyjano. Nitrile is used when -CN is a main group. The term “pyridine group” is incorrect, it is simple pyridine or pyridine ring (page 6, line 192). In scheme 13 is name benzaldehide. Putting in text phrase” All synthesized compounds were in agreement with elemental and spectral data” is unnecessary , we assume that all discussed compounds have confirmed structure and proper purity for biological tests. Similarly details of synthesis procedure must be remove from text.

In scheme 7 is mentioned reactant A but his structure is unknown. Compounds 21i-k in figure 2 contain azetidine ring.

In page 10 are mentioned four cancer lines but they are specified. It is a general that when the abbreviation is use for first time the name must be given.

In page 14 line 402 appears “lipophilic substitution” better is to use lipophilic substituent. Compounds 61 are thiosemicarbazones.

In scheme 25 number of compounds 82 and 83 does not correlate with text and structures.

Figure 8 presents intramolecular hydrogen bond.

Compound 94e exhibited MIC value, but this value is missed.

In scheme 32 appears  over arrow “Au(tht)Cl” what it means?

In page 32 is mention that ” sensitive aminoguanidines were further subjected to the minimum inhibitory concentration (MIC) test. Are known results of these measurements?

In the same page is  mentioned “saturated ammoniumthiocynate” what it means?

In page 33  voltaje-dependent and acetone must be change by voltage-dependent and acetone.

Author Response

Thanks a lot for the revision

Round 2

Reviewer 2 Report

The authors significantly improved the manuscript and took into account most of the comments of the reviewers. However, a number of problematic issues remain that should be taken into account for the final decision.

1.     Despite the authors' explanation а key remark to the peer-reviewed manuscript is the use of chemical nomenclature that is not entirely acceptable for authoritative peer-reviewed journals. In my opinion, the use of chemical names such as 4,6-disubstituted,N- -bromopropionylUBTs or 4-methyl, N-BI-N-sulfonylUBT is unacceptable. In some cases, authors may use abbreviations for staring compounds (UBT, etc), but this is unacceptable for complex structures and target derivatives. Authors should make changes throughout the text of the article by analogy with similar reviews in chemical and pharmaceutical journals. In my opinion, it is worth expanding the conclusions. In the conclusions, it is worth detailing the obtained results of literature analysis, and their possible use in further research.

2.     References list should be carefully checked and journal style policy should be strictly followed (DOI, etc). Authors should read the guide for authors and template more carefully.

My decision is still major revision.

Author Response

Reviewer 2

  1. Despite the authors' explanation а key remark to the peer-reviewed manuscript is the use of chemical nomenclature that is not entirely acceptable for authoritative peer-reviewed journals. In my opinion, the use of chemical names such as 4,6-disubstituted,N- -bromopropionylUBTs or 4-methyl, N-BI-N-sulfonylUBT is unacceptable. In some cases, authors may use abbreviations for staring compounds (UBT, etc), but this is unacceptable for complex structures and target derivatives. Authors should make changes throughout the text of the article by analogy with similar reviews in chemical and pharmaceutical journals. In my opinion, it is worth expanding the conclusions. In the conclusions, it is worth detailing the obtained results of literature analysis, and their possible use in further research.
  2. Answer: The nomenclature of (thio)ureabenzothiazole derivatives (T)UBTs was considered as proposed by some authors.

(T)UBTs = N-(substituted-BT-2-yl)-N´-(substituted)thioureas. For instance:

N-(4,6-disubstituted-BT-2-yl)-N´-(b-bromopropionyl)ureas instead of 4,6-disubstituted, N-b-bromopropionylUBTs

And N-(4-methyl-BT-2-yl)-N´-(BI-N-sulfonyl)urea instead of 4-methyl, N-BI-N-sulfonylUBT

Semicarbazide-BTs = Substituted-BT-2-yl-semicarbazides

Semicarbazone-BTs = Substituted-BT-2-yl-semicarbazones

Acyl(T)UBTs = N-(substituted-BT-2-yl)-N´-(acyloyl)(thio)ureas

GBTs = N-(substituted-BT-2-yl)-N´-substituted)-N´´-(substituted)guanidines

  1. References list should be carefully checked and journal style policy should be strictly followed (DOI, etc). Authors should read the guide for authors and template more carefully.

Answer: The references are described as indicate the guide for authors in the journal style policy

The guide for authors in the journal style policy say:

The references should be described as follows, depending on the type of work:

Journal Articles:
1. Author 1, A.B.; Author 2, C.D. Title of the article. Abbreviated Journal Name YearVolume, page range.

Reviewer 3 Report

The manuscript needs further improvements. I suggest put on the beginning of text abbreviations of all mentioned cell lines; I wary that they are not so common like suggests authors. In laboratory usage is a lot of different cancer models including the same type of cells but different by phenotype. The type writing errors appear in several places. The useless phrases should be removed. Below is a list of mistakes and necessary actions.

Page 2 line 62 is pharmacos it is not English word.

Page 3 line 111-113 remove experimental details.

Page 5 line 165 Of them?

Page 6 line 170 is: 6-Methoxy,N-methoxyphenyl-tUBTs 11a,b change to  6-Methoxy,N-(3-methoxy)phenyl-tUBTs 11a,b; the same in caption under scheme 6

Page 6 line 173 the ether cleavage of the methoxy groups, please decide ether cleavage or methoxy group cleavage.

Page 7 line 198, change afford 5-pirimidine-5- to 5-pyrimidine-5-

Page 7 line 217 is IC50  S. aureus GryB 0.25–8, please add the unit.

Page 9, scheme 8, please move reactants under the arrow.

Page 9 and 10 lines 262 – 299, in description of bioactivity of compounds 27 and 28 are not indicated their concentration indicating bioactivity.

Page 10 line, please check the name of compound 28m-q; the same in caption under Figure 3.

Page 11 line 317, is activity of 2e, is it proper number for this compound?

Page 12-line, phrase in lines 350-353 should be rebuild, in present form is misunderstanding.

Page 14, lines 384-385 is action against MES test, please change to is action under MES test conditions.

Page 16 line 439, the name d 6-substituted-BT-N-thiadiazolylsemicarbazides 56a-f, does not correspond with structure depicted in scheme 18.

Page 16 Scheme 18. The adduct obtained in addition of N-substituted semicarbazide to phenyl isothiocyanate under treatment with sulphuric acid is transformed into derivatives 56a-f, how it is possible, two carbon atoms are necessary to form 1,3,4-thiadiazole ring.

Page 17 line 463, please change 6-cloro to 6-chloro.

Page 17 line 471 separate words designed and.

Page 19 line 517 please change etyhanol to ethanol.

Page 19 line 520, the compounds 67a-6 formally are not Schiff bases but semicarbasones.

Page 19 caption under scheme 21. The compounds 68a-e are not semicarbazides. They can be named as urea derivatives.

Page 19 from phrase in lines 535 to 538 please remove experimental details.

Page 19 line 547 please specify term “substituents with nitrogen”.

Page 21 line 571, please change The Compounds to: The compounds.

Page 25 line 657, please change docked pose to Docked pose.

Page 25 scheme 26, please check structure of compounds 88a, b (valency of nitrogen atom).

Page 26 line 687 please change the dose uded to the dose used.

Page 26 line 694-695  please remove phrase with experimental details.

Page 26-line 699 change intermolecular to intramolecular and the same in caption under Figure 8.

Page 27 line 711, please change priods to periods.

Page 27 line 720 remove phrase s in stirring acetone, then refluxed for 3-4 h.

Page 27 lines 725 and 726 the concentration of applied compounds is not indicated.

Page 27 line 730 change in comparation to in comparison.

Page 27 line 731, the phrase “Also, the thiourea group connection between the BT and purine rings

seems very important for antimicrobial effect” should be rebuild.

Page 28 line 747 please change acyloylisothiocyanates by acylisothiocyanates.

Page 28 please remove experimental details, phrase “in refluxing dry toluene, stirred for 3 h”.

Page 31 line 832 please change dry pyridine to anhydrous pyridine.

Page 32 line 844 please change voltaje-dependent to voltage-dependent.

Author Response

Answer to reviewer 3

The manuscript needs further improvements.

I suggest put on the beginning of text abbreviations of all mentioned cell lines; I wary that they are not so common like suggests authors. In laboratory usage is a lot of different cancer models including the same type of cells but different by phenotype.

Answer: The abbreviations of all mentioned cell lines were put on the beginning of text

The type writing errors appear in several places. The useless phrases should be removed. Below is a list of mistakes and necessary actions.

Answer: We corrected the type writing errors and the useless phrases as well as the experimental details were removed in all text.

Page 2 line 62 is pharmacos it is not English word.

Answer: this was changed by pharmaceuticals

Page 3 line 111-113 remove experimental details.

Answer It was removed

Page 5 line 165 Of them?

Answer: It was changed by Among of them

Page 6 line 170 is: 6-Methoxy,N-methoxyphenyl-tUBTs 11a,b change to  6-Methoxy,N-(3-methoxy)phenyl-tUBTs 11a,b; the same in caption under scheme 6

Answer It was corrected.

Page 6 line 173 the ether cleavage of the methoxy groups, please decide ether cleavage or methoxy group cleavage.

Answer: It was corrected: for the methoxy group cleavage

Page 7 line 198, change afford 5-pirimidine-5- to 5-pyrimidine-5-

Answer: It was changed.

Page 7 line 217 is IC50  S. aureus GryB 0.25–8, please add the unit.

Answer: It was added.

Page 9, scheme 8, please move reactants under the arrow.

Answer: It was done

Page 9 and 10 lines 262 – 299, in description of bioactivity of compounds 27 and 28 are not indicated their concentration indicating bioactivity.

Answer: The comparative concentration bioactivity of compounds 27 and 28 were included

Page 10 line, please check the name of compound 28m-q; the same in caption under Figure 3.

Answer: IUPAC´s mane was used

Page 11 line 317, is activity of 2e, is it proper number for this compound?

Answer: 29e is the correct number

Page 12-line, phrase in lines 350-353 should be rebuild, in present form is misunderstanding.

Answer: It was rebuild.

Page 14, lines 384-385 is action against MES test, please change to is action under MES test conditions.

Answer: It was changed

Page 16 line 439, the name d 6-substituted-BT-N-thiadiazolylsemicarbazides 56a-f, does not correspond with structure depicted in scheme 18.

Answer: I am so sorry, the structure was incorrect. It was corrected.

Page 16 Scheme 18. The adduct obtained in addition of N-substituted semicarbazide to phenyl isothiocyanate under treatment with sulphuric acid is transformed into derivatives 56a-f, how it is possible, two carbon atoms are necessary to form 1,3,4-thiadiazole ring.

Answer: The structure was corrected

Page 17 line 463, please change 6-cloro to 6-chloro.

Answer: It was done

Page 17 line 471 separate words designed and.

Answer: It was done

Page 19 line 517 please change etyhanol to ethanol.

Answer: It was done

Page 19 line 520, the compounds 67a-6 formally are not Schiff bases but semicarbasones.

Answer: Were considered as semicarbazones

Reviewer 3

Page 19 caption under scheme 21. The compounds 68a-e are not semicarbazides. They can be named as urea derivatives.

Answer: It was considered as urea derivatives

Page 19 from phrase in lines 535 to 538 please remove experimental details.

Answer: This experimental details were eliminated.

Page 19 line 547 please specify term “substituents with nitrogen”.

Answer: It was considered R1-substituents with nitrogen.

Page 21 line 571, please change The Compounds to: The compounds.

----------------------

Page 25 line 657, please change docked pose to Docked pose.

Answer: It was done

Page 25 scheme 26, please check structure of compounds 88a, b (valency of nitrogen atom).

Answer: It was corrected

Page 26 line 687 please change the dose uded to the dose used.

Answer: It was corrected

Page 26 line 694-695  please remove phrase with experimental details.

Answer: The experimental details were eliminated.

Page 26-line 699 change intermolecular to intramolecular and the same in caption under Figure 8.

Answer: It was corrected

Page 27 line 711, please change priods to periods.

Answer: It was corrected

Page 27 line 720 remove phrase s in stirring acetone, then refluxed for 3-4 h.

Answer: It was removed

Page 27 lines 725 and 726 the concentration of applied compounds is not indicated.

Answer It was indicated.

Page 27 line 730 change in comparation to in comparison.

Answer: It was corrected.

Page 27 line 731, the phrase “Also, the thiourea group connection between the BT and purinerings seems very important for antimicrobial effect” should be rebuild.

Answer: It was rebuilded.

Page 28 line 747 please change acyloylisothiocyanates by acylisothiocyanates.

Answer: It was done

Page 28 please remove experimental details, phrase “in refluxing dry toluene, stirred for 3 h”.

Answer: It was removed.

Page 31 line 832 please change dry pyridine to anhydrous pyridine.

Answer: It was changed.

Page 32 line 844 please change voltaje-dependent to voltage-dependent.

Answer: It was changed

Round 3

Reviewer 2 Report

The manuscript may be accepted.

Author Response

Thank you very much for reviewing our work.

Reviewer 3 Report

The submitted new version of paper entitled” Design, synthesis, and biological activities of (thio)urea benzothiazole derivatives” has been efficiently improved. However, typewriting errors and  indicated in previous reviewer report drawback still exist.

Page 5 line 169; Caption under scheme 4  please replace “in N-(6-nitrile-BT-2-yl)-N´-(p-MeObenzyl)urea 8” by “in N-(6-cyjano-BT-2-yl)-N´-(p-MeObenzyl)urea 8.

Page 7 line 215 please change “-7-pyridie-2-yl” into “-7-pyridin-2-yl”.

Page 17 caption under scheme 18 replase dot point in name by a dash.

Page 18 caption under scheme 19; please replace “6-chloro-BT-thiosemicarbazides 57” by “6-chloro-BT-semicarbazides 57”.

Page 20 line 559; please replace “6-substituted-BT-semicazone” by “6-substituted-BT-semicarbazone”.

Page 24 line 635; please change  “N-(6-cloro-BT-2-yl)” into “N-(6-chloro-BT-2-yl)”.

Page 27 line 721 replace “intermolecular” by intramolecular.

Page 28 caption under scheme 28 please change the dash in name N-(6-substituted-BT-2-yl)N´-(benzoyl)thioureas 93a-t by shorter -.

Page 32 line 845 please explain the phrase ” with saturated ammoniumthiocynate”, saturated in what solvent?

Page 33; in scheme 38 over first arrow exists “acetona” please change by acetone.

Author Response

Answer to reviewer 3

I am sorry for the mistakes,

The typewriting errors were corrected and highlighted in green color in the text

Some other mistakes were corrected on the text

Thank you very much for reviewing our work.

The submitted new version of paper entitled” Design, synthesis, and biological activities of (thio)urea benzothiazole derivatives” has been efficiently improved. However, typewriting errors and  indicated in previous reviewer report drawback still exist.

Page 5 line 169; Caption under scheme 4  please replace “in N-(6-nitrile-BT-2-yl)-N´-(p-MeObenzyl)urea 8” by “in N-(6-cyjano-BT-2-yl)-N´-(p-MeObenzyl)urea 8.

Page 7 line 215 please change “-7-pyridie-2-yl” into “-7-pyridin-2-yl”.

Page 17 caption under scheme 18 replase dot point in name by a dash.

Page 18 caption under scheme 19; please replace “6-chloro-BT-thiosemicarbazides 57” by “6-chloro-BT-semicarbazides 57”.

Page 20 line 559; please replace “6-substituted-BT-semicazone” by “6-substituted-BT-semicarbazone”.

Page 24 line 635; please change  “N-(6-cloro-BT-2-yl)” into “N-(6-chloro-BT-2-yl)”.

Page 27 line 721 replace “intermolecular” by intramolecular.

Page 28 caption under scheme 28 please change the dash in name N-(6-substituted-BT-2-yl)—N´-(benzoyl)thioureas 93a-t by shorter -.

Page 32 line 845 please explain the phrase ” with saturated ammoniumthiocynate”, saturated in what solvent?

The article says.

with saturated ammoniumthiocynate and concentrated hydrochloric acid

Page 33; in scheme 38 over first arrow exists “acetona” please change by acetone.